# Antimicrobial Activity and Proposed Action Mechanism of 3-Carene against *Brochothrix thermosphacta* and *Pseudomonas fluorescens*

**DOI:** 10.3390/molecules24183246

**Published:** 2019-09-06

**Authors:** Huizhen Shu, Haiming Chen, Xiaolong Wang, Yueying Hu, Yonghuan Yun, Qiuping Zhong, Weijun Chen, Wenxue Chen

**Affiliations:** 1College of Food Sciences & Engineering, Hainan University, 58 People Road, Haikou 570228, China (H.S.) (H.C.) (X.W.) (Y.H.) (Y.Y.) (Q.Z.); 2Chunguang Agro-Product Processing Institute, Wenchang 571333, China

**Keywords:** 3-carene, antibacterial mechanism, membrane damage, ATP, TCA cycle, metabolic disorder, genomic DNA

## Abstract

3-Carene is an antimicrobial monoterpene that occurs naturally in a variety of plants and has an ambiguous antibacterial mechanism against food-borne germs. The antibacterial effects and action mechanism of 3-carene against Gram-positive *Brochothrix thermosphacta* ACCC 03870 and Gram-negative *Pseudomonas fluorescens* ATCC 13525 were studied. Scanning electron microscopy (SEM) examination and leakage of alkaline phosphatase (AKP) verified that 3-carene caused more obvious damage to the morphology and wall structure of *B. thermosphacta* than *P. fluorescens*. The release of potassium ions and proteins, the reduction in membrane potential (MP), and fluorescein diacetate (FDA) staining further confirmed that the loss of the barrier function of the cell membrane and the leakage of cytoplasmic contents were due to the 3-carene treatment. Furthermore, the disorder of succinate dehydrogenase (SDH), malate dehydrogenase (MDH), pyruvate kinase (PK), and ATP content indicated that 3-carene could lead to metabolic dysfunction and inhibit energy synthesis. In addition, the results from the fluorescence analysis revealed that 3-carene could probably bind to bacterial DNA and affect the conformation and structure of genomic DNA. These results revealed that 3-carene had strong antibacterial activity against *B. thermosphacta* and *P. fluorescens* via membrane damage, bacterial metabolic perturbations, and genomic DNA structure disruption, interfering in cellular functions and even causing cell death.

## 1. Introduction

Food spoilage is caused by microorganisms that produce unpleasant sensory effects through various metabolic activities [1]. Food contamination caused by facultative pathogens is a serious challenge in the food industry [2]. *Brochothrix thermosphacta* are Gram-positive and facultative anaerobic bacteria that can cause spoilage in chilled meat and seafood products [3]. Jiang et al. [4] showed that *B. thermosphacta* are the dominant spoilage bacteria in vacuum-packed pork during chilled storage due to the fact of its facultative anaerobic and psychrophilic features, leading to corruption and deterioration. *Pseudomonas fluorescens* are Gram-negative bactera and are commonly found in cold meat, seafood, and dairy products and can cause serious diseases such as septicemia, septic shock, and intravascular coagulation [5,6,7]. Various strategies have been developed to inhibit the emergence and spread of pathogenic and spoilage bacteria in food for decades, such as the use of chemical preservatives. However, long-term and excessive addition of such chemical preservatives creates a potential risk to public health [8,9]. The improper use of chemical preservatives may cause problems such as carcinogenicity, teratogenicity, and residual toxicity [10]. In fact, natural antibacterial agents can be applied as a food preservative instead of the currently popular chemical preservatives [11,12]. Therefore, searching for safe and natural antibacterial agents in the future, including plant extracts and essential oils (EOs), is a crucial priority.

Essential oils are usually extracted from spices and herbs because of their good sensory and antiseptic properties, and EOs have become effective natural preservatives to inhibit microbial growth, extend shelf life, and benefit human health [13,14]. Our group reported some antibacterial activities and mechanisms of EOs extracted from spices and plants. For example, black pepper chloroform extract can inhibit the growth of *Escherichia coli* and *Staphylococcus aureus* by disrupting the membrane structure, leading to energy synthesis and metabolic dysfunction [15]; black pepper petroleum ether extract showed good inhibitory effect on *Listeria monocytogenes* and *Salmonella Typhimurium* [16]. Monoterpenes are the main component of EOs and display various biological activities, including antibacterial activity [17]. Thymol, (+) menthol and linalyl acetate are three common monoterpenes that have effective antibacterial activity against the Gram-positive bacterium *S. aureus* and Gram-negative bacterium *E. coli* via destroyed cell walls and membranes and causing leakage of intracellular substances [18]. 3,7,7-Trimethyl-bicyclo [4,1,0]hept-3-ene (3-carene) is the main components of some conifers’ and herbs’ volatile oils, such as *Pinus* and pepper. 3-Carene is an effective natural inhibitor of food-borne germs with high contents of black and white pepper oils [19]. However, the potential inhibition mechanism of 3-carene is not fully understood.

The inhibition mechanism of antibacterial compounds has been explored by examining cell wall and membrane permeability damage, protein changes, and changes in enzyme activity and nucleic acid synthesis, etc. [20]. In this study, the antimicrobial activity and action mechanism of 3-carene against the Gram-positive *B. thermosphacta* and Gram-negative *P. fluorescens* were investigated using a series of behavioral and biochemical methods. Scanning electron microscopy (SEM) was employed to observe the cell damage and dynamic morphological changes in the presence of 3-carene. The leakage of cell contents, various enzyme activities, and ATP concentration were determined to explore the antibacterial mechanism of 3-carene. Finally, the effect of 3-carene on bacterial genomic DNA was also studied by fluorescence spectroscopy.

## 2. Results

### 2.1. Inhibition Effect of 3-Carene on B. thermosphacta and P. fluorescens Strains

The minimum inhibitory concentration (MIC) of 3-carene against *B. thermosphacta* and *P. fluorescens* was determined by the agar dilution method. As shown in Table 1, the MIC of 3-carene against both strains was 20 mL/L, revealing the intense antibacterial activities of 3-carene against *B. thermosphacta* and *P. fluorescens*.

As shown in Figure 1, bacterial growth in the control groups followed the model s-shaped growth curve in general. *Brochothrix thermosphacta* and *P. fluorescens* reached the logarithmic phase after 7 and 5 h, respectively, and then stationary phase after 17 h. When 3-carene was added to the medium, *B. thermosphacta* stopped growing (Figure 1a) and *P. fluorescens* grew slowly (Figure 1b). The bacteria grew slowly when 3-carene was added to the medium. When grown in media containing 3-carene at 1× MIC and 2× MIC, the OD600 of *B. thermosphacta* and *P. fluorescens* was critically lower than the control groups in each growth phase.

### 2.2. Cell Wall Permeability of B. thermosphacta and P. fluorescens

Alkaline phosphatase (AKP) is a kind of intracellular enzyme that is found in many prokaryotes and localizes between the cell wall and membrane [21]. Under normal conditions, AKP activity is not detected extracellularly unless the cell wall is destroyed and AKP leaks into the extracellular environment [22]. As shown in Figure 2a, the extracellular AKP activity of *B. thermosphacta* in the absence of 3-carene (blank control group and negative control group) was maintained at approximately 0.408 U. After treatment with 3-carene (1× MIC and 2× MIC), the AKP activity increased with growth and time, until 8 h when the activity increased to 4.56 and 8.04 U, almost 11 fold and 20 fold of the control groups, respectively. It can clearly be observed that the AKP activity of *B. thermosphacta* increased as the 3-carene concentration increased from 1× MIC to 2× MIC. However, there was no significant difference in the extracellular AKP activity of *P. fluorescens* treated with 3-carene compared to the control group (Figure 2b).

### 2.3. Release of Potassium Ion

Determining the efflux of potassium ions in bacterial cells is a classic method to study the damage of cell membranes by antibacterial drugs [23]. The effect of 3-carene on the cell wall and membrane of *B. thermosphacta* and *P. fluorescens* was further explored by potassium ion release assay. Figure 2c indicated that the presence of potassium ions was not detected in the *B. thermosphacta* control groups. The addition of 3-carene caused a dramatic increase in potassium ion release from 0–0.5 h and a relatively stable state at 0.5–3 h, then an upward trend from 3–4 h. For *P. fluorescens* (Figure 2d), the control groups showed a slight increase at 0–1 h and then stabilized. With the addition of 3-carene, the potassium ion release increased sharply from 0 to 0.5 h.

### 2.4. The Effect of 3-Carene on Inhibiting Cell Membrane of B. thermosphacta and P. fluorescens

Fluorescein diacetate (FDA), an uncharged lipid molecule that does not fluoresce, can easily pass through the cell membrane where it is hydrolyzed into fluorescein by an enzyme and remains in the cell, emitting yellow-green fluorescence due to the polarity of fluorescein [24]. The loss of fluorescein molecules leads to a decrease in FDA fluorescence intensity, while the cell membrane is destroyed [25]. Therefore, the integrity and permeability of bacterial cell membranes can be reflected by changes in FDA fluorescence intensity. The fluorescein remaining on the cell membrane may be up-taken by FDA and then undergo intracellular hydrolysis by esterase to release free fluorescein [26]. Therefore, this method can also be used as an indicator of cell viability and metabolic activity of bacteria [27]. As shown in Figure 3a, the fluorescence intensity of untreated *B. thermosphacta* increased over time, while with the addition of 3-carene (1× MIC and 2× MIC), the fluorescence intensity significantly decreased almost 7 fold in the control groups after 12 h, indicating that the structure of the cell membrane in *B. thermosphacta* was destroyed by 3-carene. For *P. fluorescens* (Figure 3b), there was little diversification in the fluorescence intensity of the control groups, while a significant decrease (1 fold and 2 fold the control groups) occurred when the cells were treated with 3-carene (1× MIC and 2× MIC). Moreover, a greater reduction in fluorescence intensity was observed along with an increase in the concentration of 3-carene from 1× MIC to 2× MIC, indicating that the serious damage of the *P. fluorescens* cell membrane may be caused by the increase in the 3-carene concentration.

### 2.5. Effect of 3-Carene on the Cell Ultrastructure of B. thermosphacta and P. fluorescens

To evaluate the changes in membrane damage and cellular structure of *B. thermosphacta* and *P. fluorescens* in response to 3-carene, SEM measurements were employed (Figure 4). Scanning electron microscopy clearly showed that most cells in the control groups (blank control and negative control) (Figure 4a–d,i–l) were intact and smooth with uniform cytoplasmic appearance and clear cell boundaries without notable rupture pores. After treatment with 3-carene at 1× MIC and 2× MIC for 4 and 8 h, the morphological damage of *B. thermosphacta* was severe (Figure 4e–h), the number of abnormal cells increased, and stomatal formation and cell lysis were obviously increased. After exposure of *P. fluorescens* to 3-carene (3-carene at 1× MIC and 2× MIC for 4 and 8 h) (Figure 4m–p), the cell distortion increased and included cells that were swollen, dumb-bell-shaped, and had bulbous ends. Moreover, disintegration of *P. fluorescens* cells and debris near the treated cells were also observed.

### 2.6. Protein Synthesis

The effect of 3-carene on the intracellular protein contents of *B. thermosphacta* and *P. fluorescens* is shown in Figure 5. The protein contents of both untreated cells increased with the growth of bacteria and reached a maximum value after incubation for 24 h. The intracellular protein concentration of 3-carene-treated *B. thermosphacta* significantly decreased (*p* < 0.05) (Figure 5a). The variation in protein contents of *P. fluorescens* with the addition of 3-carene was similar to the control groups in 0–12 h. The protein contents of treated groups (1× MIC and 2× MIC) were 0.45 and 0.37 mg/mL, whereas the content of the control groups was almost 0.65 mg/mL after incubation for 24 h (Figure 5b).

### 2.7. Effect of 3-Carene on Bacterial Membrane Potential (MP)

The results of the bacterial MP analysis were expressed as the mean fluorescence intensity (MFI) of Rhodamine 123 (Figure 6). With the treatment of 3-carene at 1 × MIC, a dramatic decline appeared in both *B. thermosphacta* and *P. fluorescens*, with the MFI values decreasing to almost 34% and 61% compared to the control groups, respectively. Moreover, the MFI of *B. thermosphacta* and *P. fluorescens* with the addition of 3-carene at 2× MIC decreased by 41% and 80%, respectively. In addition, it also revealed that the MFI values of *P. fluorescens* were lower than those of *B. thermosphacta* when treated with 3-carene, and the MP decreased while 3-carene concentration increased.

### 2.8. Effect of 3-Carene on Enzyme Activity

The effects of 3-carene on the succinate dehydrogenase (SDH) and malate dehydrogenase (MDH) activities in *B. thermosphacta* are shown in Figure 7. The SDH and MDH activities of both bacteria generally fluctuated, while the values of the untreated groups were significantly higher than the treated groups. The SDH value decreased sharply after treatment with 3-carene at 1× MIC and 2× MIC for 3 h and reached minimum values of 13.95 and 12.56 U/mg prot, respectively (Figure 7a). The SDH values of the control groups were 46.33 and 46.99 U/mg prot when incubated with 3-carene at 1× MIC and 2× MIC for 24 h, respectively, while the treated groups were 20.86 and 20.97 U/mg prot. As shown in Figure 7b, the MDH activity in the blank control group showed a fluctuating change within 24 h, and the negative control group showed a downward trend at 3 h. The MDH activity decreased exponentially after treatment with 3-carene until its value was merely 0.02 in 24 h.

The effects of 3-carene on the SDH and MDH activities in *P. fluorescens* are shown in Figure 7. The SDH activity in *P. fluorescens* of the control groups increased slightly at 0–3 h and then decreased for 3–12 h until it rose slightly at 24 h (Figure 7c). The SDH activity of the treated groups had a similar pattern to that of the control groups. The SDH activity of the 2× MIC group was significantly lower than that of the 1× MIC group at 12 and 24 h. The MDH activity of the control groups in *P. fluorescens* attained a maximum when treated with 3-carene for 3 h. The MDH of *P. fluorescens* treated with 3-carene was much lower than that of the control groups (Figure 7d).

As shown in Figure 7e,f, the activity of pyruvate kinase (PK) for *B. thermosphacta* and *P. fluorescens* dramatically increased in the control groups. *B. thermosphacta* with the addition of 3-carene first increased slightly and then decreased. *P. fluorescens* treated with 3-carene at 1× MIC significantly increased during 1–9 h and decreased during 9–4 h. While the inhibitory concentration was 2× MIC, the PK activity started to become inhibited after treatment for 6 h.

### 2.9. Effect of 3-Carene on the Bacterial ATP Concentration

The effect of 3-carene on the ATP concentration of *B. thermosphacta* and *P. fluorescens* is shown in Figure 8. In the untreated *B. thermosphacta* and *P. fluorescens*, there was a slight increase in ATP concentrations due to the cell growth and then a slight decrease. With the treatment of 3-carene at 1 × MIC, the ATP concentration of *B. thermosphacta* decreased sharply from 0–10 h and then reached a steady state, while after treatment with 3-carene at 2× MIC, the ATP concentration of the cells decreased sharply within 3 h (Figure 8a). For *P. fluorescens*, the addition of 3-carene resulted in a decrease in ATP concentration from 0 to 24 h (Figure 8b).

### 2.10. Effect of 3-Carene on B. thermosphacta and P. fluorescens DNA

The interaction between 3-carene and DNA was investigated using fluorescence spectroscopy [28,29]. As shown in Figure 9, the fluorescence quenching increased significantly with the increase in 3-carene concentration, which may be because 3-carene could probably bind to *B. thermosphacta* and *P. fluorescens* DNA, leading to an alteration in the conformation of the DNA.

## 3. Discussion

This research explored the antibacterial activity and possibly inhibition mechanism of 3-carene against Gram-positive *B. thermosphacta* ACCC 03,870 and Gram-negative *P. fluorescens* ATCC 13525. Lizandra et al. [30] identified that 3-carene was the main chemical constituent of essential oil of *Piper cubeba* and significant cytotoxicity was only obtained in the concentration of 200 μg/mL after 24 h treatment. The results of MIC indicated that 3-carene showed remarkable inhibitory effect on *B. thermosphacta* and *P. fluorescens*. The growth curve revealed that 3-carene could effectively delay the growth of *B. thermosphacta* and *P. fluorescens*.

The cell membrane acts as a barrier to bacterial life and maintains a balance of materials and energy [31]. The SEM observation illustrated that with the treatment of 3-carene, the cell morphologies of both bacteria were destroyed to varying degrees. Yao et al. [32] showed that serious plasmolysis was evident, confirming the severely damaged cell membrane structures of *P. fluorescens* and *P. aeruginosa* treated with polymethoxylated flavone. A similar report revealed that different lesions appeared in oregano and clove EO-treated cells; holes in the cell wall of *E. coli* and damages such as cell deformity in *B. subtilis* were observed [33]. The cell wall and membrane structure of *B. thermosphacta* were severely injured, while *P. fluorescens* suffered less damage, which is consistent with the results of AKP and FDA. The results indicated that 3-carene may first destroy the cell wall integrity of *B. thermosphacta*, resulting in the leakage of AKP. The destruction of the *P. fluorescens* cell wall by 3-carene only caused the distortion of its morphology instead of cell wall rupture. Low FDA fluorescence intensity indicated disruptions in the cell membrane integrity of the cell, and fluorescence leaked from the cytosol. The lower leakage of AKP and FDA molecules in *B. thermosphacta* indicated that more serious damage to the cell membrane occurred in *B. thermosphacta* than in *P. fluorescens* when treated with 3-carene.

The microbial cell membrane maintains the balance of intracellular material and energy, allowing cells to maintain normal life activities, and the target of most antimicrobial drugs is the cell membrane [32]. Researchers believed that the first sign of cell damage was the outflow of K^+^ and Na^+^, followed by the release of cellular contents [34]. An increase in the amount of potassium ion efflux from the cells of *B. thermosphacta* and *P. fluorescens* provided further evidence that 3-carene causes damage to the cell wall and membrane, leading to leakage of protoplasmic content [35].

We observed a decrease of protein content in *B. thermosphacta* and *P. fluorescens* with the addition of 3-carene. Similarly, it was found that rosmarinic acid could affect the cell membrane permeability of *E. coli* and *S. aureus*, causing the reduction of proteins [36]. Evidence on the leakage of cellular components indicates the integrity of the cell membrane. Our research revealed that intracellular substances, including potassium ions and proteins, leaked into the cell suspension after the addition of 3-carene. Therefore, it was supposed that 3-carene could result in protein leakage and disrupt protein synthesis, cause metabolic dysfunction, and even lead to cell death. Furthermore, the inhibition ability of 3-carene in *B. thermosphacta* was stronger than that in *P. fluorescens*.

Membrane potential (MP) plays an important role in bacterial physiology, which can well explain the antibacterial mechanism of antibacterial substances [37]. As an element of proton dynamics, MP participates in the generation of ATP, and its changes can affect cellular metabolic activity [38]. The MP of normal bacteria is generated by the difference in the intracellular and extracellular ion concentrations. The MP of bacteria is produced by the divergence in the intracellular and extracellular ion concentrations inside and outside the cell membrane, and if the cell is depolarized, the value of MP will decrease [39]. Reference [40] reported that three surfactants interfere with the normal growth of *S. typhimurium* and *S. aureus* due to the alterations in the bacterial structure and MP. The results of this study revealed that the bacterial average fluorescence intensity of R 123 was significantly decreased with the treatment of 3-carene, suggesting that cell membrane depolarization caused irregular cellular metabolic activity and bacterial death.

The TCA cycle is the major central pathway linking the great majority of individual metabolic pathways and is an essential metabolic pathway for all aerobic organisms [41]. Adenosine triphosphate (ATP) is related to a massive number of cellular functions, energy metabolism and life activities and is an important indicator for exploring the mechanism of inhibition [42]. As shown in Figure 10, ATP is involved in multiple links in the TCA cycle and the glycolysis pathway. In addition, the consumption of intracellular ATP is related to changes in membrane potential, which corresponds to the results of the MP analysis. It has been suggested that the decrease in ATP concentration might be due to the decrease in the rate of intracellular ATP synthesis and the increase in the rate of ATP hydrolysis, as well as severe damage to the cell membrane, allowing inorganic phosphate to shuttle freely [43]. The results revealed that 3-carene could probably destroy or even kill the bacteria and cause a rapid degradation of ATP. The severe decrease in bacterial ATP concentration with the addition of 3-carene may be due to the inhibition of ATP synthesis, the increase in ATP hydrolysis and biofilm structure interruption, confirming the damage of bacterial energy metabolism and biosynthetic pathways. This may lead to TCA cycle defects due to TCA cycle flow interruption and energy production damage.

Antibacterial agents can be judged by the damage to cell membrane structure and the disorder of enzyme system function [44]. Leakage and activity disorder of intracellular enzymes could reveal that cells suffer from membrane damage and metabolic dysregulation [45]. As show in Figure 10, both SDH and MDH are key enzymes in eukaryotic and prokaryotic cells and are involved in the TCA pathway, playing an essential role in cellular energy metabolism, and the activity of SDH and MDH reflects the energy metabolic status of cells [46]. As a part of the respiratory chain, SDH is the only multi-subunit enzyme involved in oxidative phosphorylation in the cell membrane and is one of the hubs of electron transport in the TCA cycle [47]. Succinate is formed from the conversion of succinyl coenzyme A and then oxidized to fumarate by SDH or complex II of the electron transport chain, transferring electrons to power ATP synthase [46]. As a coenzyme of NADP+, MDH receives hydrogen from metabolites during biosynthesis to form NADPH [48]. PK promotes the production of pyruvate in the glycolytic pathway and promotes energy metabolism. Pyruvate is the raw material of acetyl-CoA, which is the starting point for the TCA cycle [49]. Overall, the activities of SDH, MDH, and PK in the treated groups were lower than those of the control groups during the examined incubation period. The decrease in enzyme activities after 3-carene treatment might be due to the structural change imposed by a reaction with the side chain of the enzyme, leading to a conformational change in the enzymes. In addition, the bacterial respiratory chain is located on the cell membrane, so the cell membrane is involved in various important life activities of bacteria, such as cell energy conversion, structural macromolecule synthesis, and enzyme secretion [50]. Consequently, 3-carene can interfere with bacterial respiration and energy metabolism by disrupting the membrane structure and the enzyme system function in the respiratory chain, thereby disrupting the normal life activities of the bacteria.

The destruction of genetic material, DNA, could impede the expression of genes, thereby blocking the synthesis of normal enzymes and receptors and even leading to bacterial death [51]. Mori et al. [52] showed that flavonoids can effectively inhibit DNA synthesis of *Proteus vulgaris*. In this study, it was found that the bacterial genome might be an antibacterial target of 3-carene, and 3-carene could disrupt gene expression in addition to the cell wall, cell membrane, and metabolic processes, which enriched the understanding of the antibacterial mechanism of 3-carene.

## 4. Materials and Methods

### 4.1. Bacterial Strains and Chemicals

The Gram-positive bacterium *B. thermosphacta* ACCC 03,870 was purchased from the Agricultural Culture Collection of China, and the Gram-negative bacterium *P. fluorescens* ATCC 13,525 was obtained from the Guangdong Culture Collection Center. The strains were stored in sterile slants of nutrient agar (NA) at 4 °C. The (+)-3-Carene was purchased from Tokyo Chemical Industry Co., Ltd. (Tokyo, Japan). The SDH, MDH, PK, and ATP assay kits were purchased from Nanjing JianCheng Bioengineering Institute (Nanjing, China). Rhodamine 123 was purchased from Shanghai Yuanye Bio-Technology Co., Ltd. (Shanghai, China). All other chemicals were of analytical grade unless otherwise mentioned.

### 4.2. Determination of MIC

The MIC of 3-carene was tested using an agar dilution method [53]. Briefly, a two-fold serial dilution method was used and 3-carene was dissolved in alcohol (20%, *v*/*v*) to obtain a concentration of 400 mL/L and serially diluted to 200, 100, 50, 25, 12.5, and 6.25 mL/L. A prepared test compound solution (2 mL) was mixed with nutrient agar medium (18 mL) to yield final concentrations of 40, 20, 10, 5, 2.5, 1.25 and 0.625 mL/L. The final concentration of ethanol in the culture medium was maintained at 2% (*v*/*v*). Sterile water was added to bacterial suspensions as a blank control and ethanol (2%, *v*/*v*) (3-carene solvent) was added to bacterial suspensions as a negative control to check their normal growth. All experiments in this study used this method as a blank and negative control. A 200 μL standardized bacterial suspension containing 10^6^–10^7^ CFU/mL of the bacterial was transferred to each plate. After inoculation, *B. thermosphacta* and *P. fluorescens* were incubated at 26 °C and 30 °C for 24 h, respectively. A plate without bacterial growth was visually observed by comparing it with the control groups without drug growth. The MIC expressed in mL/L was defined as the lowest concentration at which 3-carene completely inhibited bacterial growth.

### 4.3. The Inhibition Effect of 3-Carene on the Growth Curve

The inhibition activities of 3-carene on the growth of *B. thermosphacta* and *P. fluorescens* were determined by the ultraviolet spectrophotometry method [54]. Briefly, *B. thermosphacta* ACCC 03,870 and *P. fluorescens* ATCC 13,525 were grown to approximately 10^6^–10^7^ CFU/mL in nutrient broth (NB) and inoculated into fresh NB medium to a final density of 5%. 3-Carene was dissolved in ethanol (20%, *v*/*v*) and added to the cultures with a 10% addition amount to obtain final concentrations of 1× MIC and 2× MIC. The mixture was cultured at 26 °C and 30 °C with stirring (180 rpm/min), and the cell concentration was determined by measuring the OD600 nm at 1 h intervals by a UV spectrophotometer (TU1810, Beijing Purkinje General Instrument Co., Ltd., Beijing, China). A bacterial growth curve was determined to analyze the activities of 3-carene on *B. thermosphacta* and *P. fluorescens* growth.

### 4.4. Determination of Cell Wall Integrity

The changes in cell wall permeability were evaluated by measuring the AKP activity [55]. Briefly, 3-carene was added to the logarithmic phase bacteria suspension (1 × 10^6^–10^7^) to obtain the final concentrations of 1× MIC and 2× MIC and incubated at 26 °C and 30 °C, respectively. After incubation for 6, 9, and 12 h, the suspension was collected by centrifugation, and the cellular AKP activity was determined using an AKP kit with a UV spectrophotometer.

### 4.5. Potassium Ion Release Assay

The potassium release assay was employed to determine the cell membrane integrity of *B. thermosphacta* and *P. fluorescens* treated with 3-carene. The exponential phase cells were centrifuged, washed and resuspended in 0.9% sterile saline (10^6^~10^7^ CFU/mL). The cell suspensions were incubated with 3-carene (final concentration 1× MIC and 2× MIC) at 30 °C. At multiple time intervals (0, 0.5, 1, 2, 3, and 4 h), the cell suspensions were centrifuged at 10,000× *g* for 10 min, and then the supernatants were measured using an atomic absorption spectrometer (TAS-990 Super AFG, Beijing Purkinje General Instrument Co., Ltd, Beijing, China) [56]. All assays were carried out in triplicate.

### 4.6. Fluorescein Diacetate Staining Experiment

The effect of 3-carene on the membrane permeability of *B. thermosphacta* and *P. fluorescens* was studied by fluorescein diacetate (FDA) staining [57,58]. Suspensions of the logarithmic growth phase cells were centrifuged, and the precipitation was suspended in physiological saline adjusted to 10^6^–10^7^ CFU/mL. A solution of 3-carene (1× MIC and 2× MIC) was added to the culture and incubated on an orbital shaker at 180 rev/min at the optimum growth temperature. After incubation for 6, 9, and 12 h, cells were harvested by centrifugation at 4000× *g* for 10 min and washed twice with PBS (pH 7.4). Then, 250 μL of FDA acetone solution (2 mg/mL) was added to the precipitate, which was incubated for 20 min, washed 3 times with PBS (pH 7.4), and then resuspended in PBS. The fluorescence of the mixture was measured by a fluorescence spectrophotometer (WGY-10, Tianjin Gangdong Sci. and Tech. Development Co. LTD, Tianjin, China) with excitation and emission wavelengths of 297 and 527 nm, respectively; the slit width was 10 nm.

### 4.7. Scanning Electron Microscopy (SEM) Observation

The SEM method was used for observing bacterial cell morphology [59]. Suspensions of the logarithmic growth phase cells were treated with 3-carene at 1× MIC and 2× MIC. After incubation for 4 and 8 h, cells were harvested by centrifugation and washed three times with PBS (pH 7.4). Then, the cells were dehydrated in water–alcohol solutions at different alcohol concentrations (20%, 40%, 60%, 80%, and 100%) three times. The dehydrated samples were pre-freezed at −20 °C for 2 h and freeze-dried for 12 h. The samples were then sputter-coated with gold under vacuum and examined by SEM (S-4800, Hitachi, Tokyo, Japan).

### 4.8. Membrane Potential (MP) Determinations

To analyze the effects of 3-carene on the metabolic activity of *B. thermosphacta* and *P. fluorescens*, the MP of bacteria was determined by the Rhodamine fluorescence method [60]. Different concentrations of 3-carene (1× MIC and 2× MIC) were added to the logarithmic growth phase bacteria and incubated for 3 h. The bacteria were washed twice with PBS and 1 mg/mL rhodamine 123 stock solution was added to obtain a final concentration of 2 μg/mL. The samples were incubated in the dark for 30 min, washed 3 times, and resuspended in PBS. The mean fluorescence intensity (MFI) of the cell suspension was determined by excitation and emission wavelengths at 480 and 530 nm, respectively, in a fluorescence spectrophotometer.

### 4.9. Protein Content, and SDH, MDH, and PK Activities

Bacterial cells were cultured to the logarithmic growth phase (approximately 10^6^–10^7^ CFU/ml) and treated with various concentrations of 3-carene (1× MIC and 2× MIC). After incubation for 0, 3, 6, 9, 12, and 24 h, the cells were collected, washed completely, resuspended in PBS, and then disrupted by ultrasonic treatment (550 W, working for 3 s at 5 s intervals) for 5 min in an ice bath. The cell debris was removed by centrifugation, and the supernatant was stored at 4 °C for protein, SDH, MDH, and PK determinations. The activities of SDH, MDH, and PK were expressed based on the protein concentration which determined by the Coomassie Brilliant Blue method [61]. Three milliliters of Bio-Rad assay solution was added to 100 μL of bacterial supernatant and incubated for 10 min. The absorption at 595 nm was measured against a blank. SDH, MDH, and PK activities were assayed using the SDH and MDH assay kits (JianCheng Bioengineering Institute, Nanjing, China) according to their operation instructions.

### 4.10. Measurement of Intracellular ATP Concentrations

The broken cell supernatant was obtained by the procedure similar to 2.8, but the cells were washed and suspended in sterile saline instead of PBS [62]. The intracellular ATP concentration was measured by ATP assay kit (JianCheng Bioengineering Institute, Nanjing, China), according to the manufacturer’s instructions. Then, the reacted mixture was transferred to a 96 well microtiter plate and monitored using a multimode plate reader (SP-Max3500FL, Shanghai Flash Spectrum Biotechnology Co., Ltd., Shanghai, China) at 636 nm.

### 4.11. The Effect of 3-Carene on Bacterial Genomic DNA

A TIANamp Bacteria DNA Kit (Tiangen Biotech, Co., Ltd., Beijing, China) was employed to abstract the genomic DNA of *B. thermosphacta* and *P. fluorescens* based on the manufacturer’s instructions. The purity and concentration of the extracted DNA were determined by the optical density ratio of 260 and 280 nm (OD 260/OD 280 = 1.8) and OD260 nm, respectively, on an ultramicro spectrophotometer (NanoPhotometer-N50, Implen, Germany) [56]. The effect of 3-carene on DNA was determined with a multifunctional fluorescence microplate reader (SP-Max3500FL). Bacterial genomic DNA was diluted to 60 μg/mL with 0.01 M Tris-HCl (pH 7.2) buffer. Different concentrations of 3-carene solution (0, MIC, 2 MIC, 4 MIC) were added to the same volume of the 60 μg/mL DNA solution and incubated for 10 min at 37 °C in the dark. The excitation was set at 280 nm, and the fluorescence spectra of the mixture were measured from 300 to 500 nm. The slit width was 10 nm.

### 4.12. Statistical Analysis

All experiments in this study were executed in triplicate, and the mean value was calculated with standard deviation. Statistical analysis was performed using SPSS version 13.0 statistical software (SPSS Inc., Chicago, IL, USA) for analysis of variance. Graphs were created by Origin software (Origin Lab Co., Pro.8.0, Northampton, MA, USA). Differences among groups were considered significant when *p* < 0.05.

## 5. Conclusions

In summary, 3-carene showed effective inhibitory activity on Gram-positive *B. thermosphacta* and Gram-negative *P. fluorescens*. The results revealed that 3-carene could delay the growth of bacteria and even lead to cell death. 3-Carene could destroy the normal morphology and life activities of the cell wall and membrane, leading to the leakage of macromolecular substances, such as intracellular proteins and potassium ions. 3-Carene could inhibit the activity of the intracellular tricarboxylic acid cycle and glycolysis-associated enzymes and disrupt the synthesis and decomposition of ATP, causing bacterial metabolic dysfunction. In addition, 3-carene was found to bind to DNA and interfere with gene expression. In short, the antibacterial mode of action of 3-carene can be attributed to multiple pathways but is primarily associated with cell membrane integrity and cell damage. Thus, 3-carene is expected to be a candidate as an effective natural inhibitory compound and may be evolved as a multifunctional food additive. We will continue this research direction to demonstrate the antimicrobial activity and inhibition mechanism of monoterpene using advanced experimental techniques to ensure that natural inhibitors can be safely and effectively applied to the food industry.

## Figures and Tables

**Figure 1 molecules-24-03246-f001:**
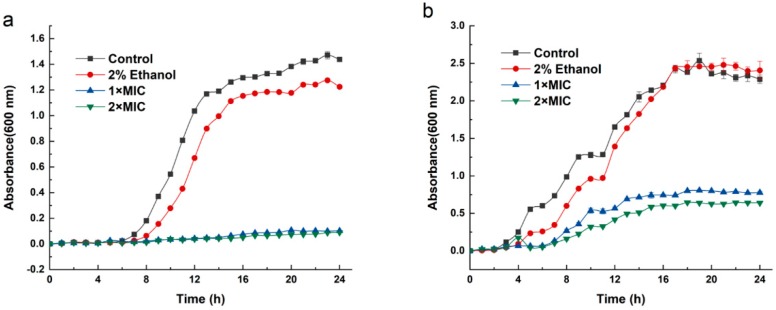
Effect of different 3-carene concentrations on bacterial dynamic growth curves. (**a**) *B. thermosphacta*, (**b**) *P. fluorescens*.

**Figure 2 molecules-24-03246-f002:**
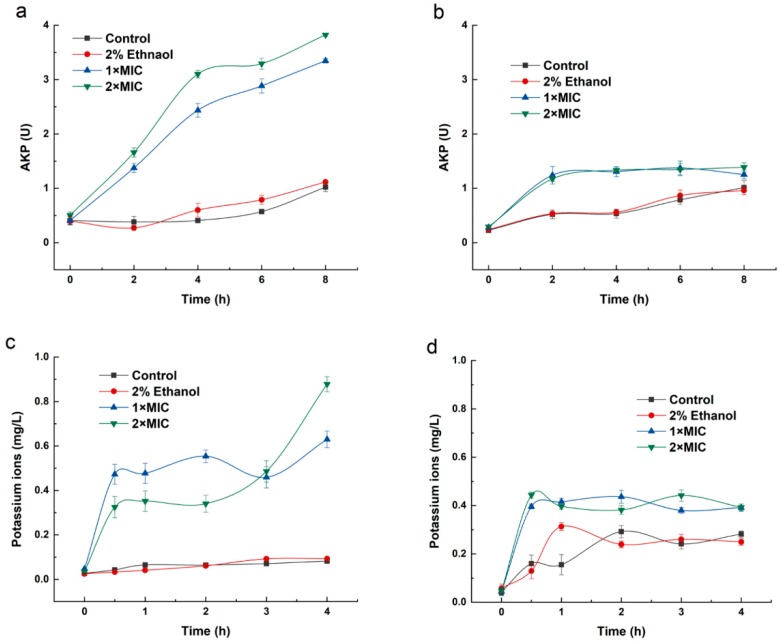
Alkaline phosphatase (AKP) activity of bacteria before and after 3-carene treatment: (**a**) *B. thermosphacta*, (**b**) *P. fluorescens*. Relative amount of potassium ions released by bacteria treated with 3-carene: (**c**) *B. thermosphacta*, (**d**) *P. fluorescens*.

**Figure 3 molecules-24-03246-f003:**
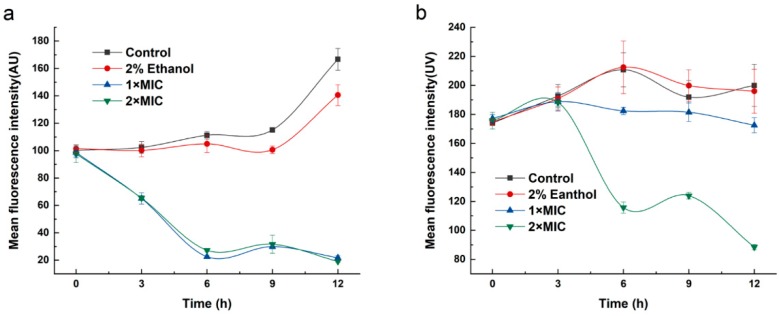
Effect of 3-carene on the fluorescein diacetate (FDA) fluorescence intensity of bacteria: (**a**) *B. thermosphacta*, (**b**) *P. fluorescens*.

**Figure 4 molecules-24-03246-f004:**
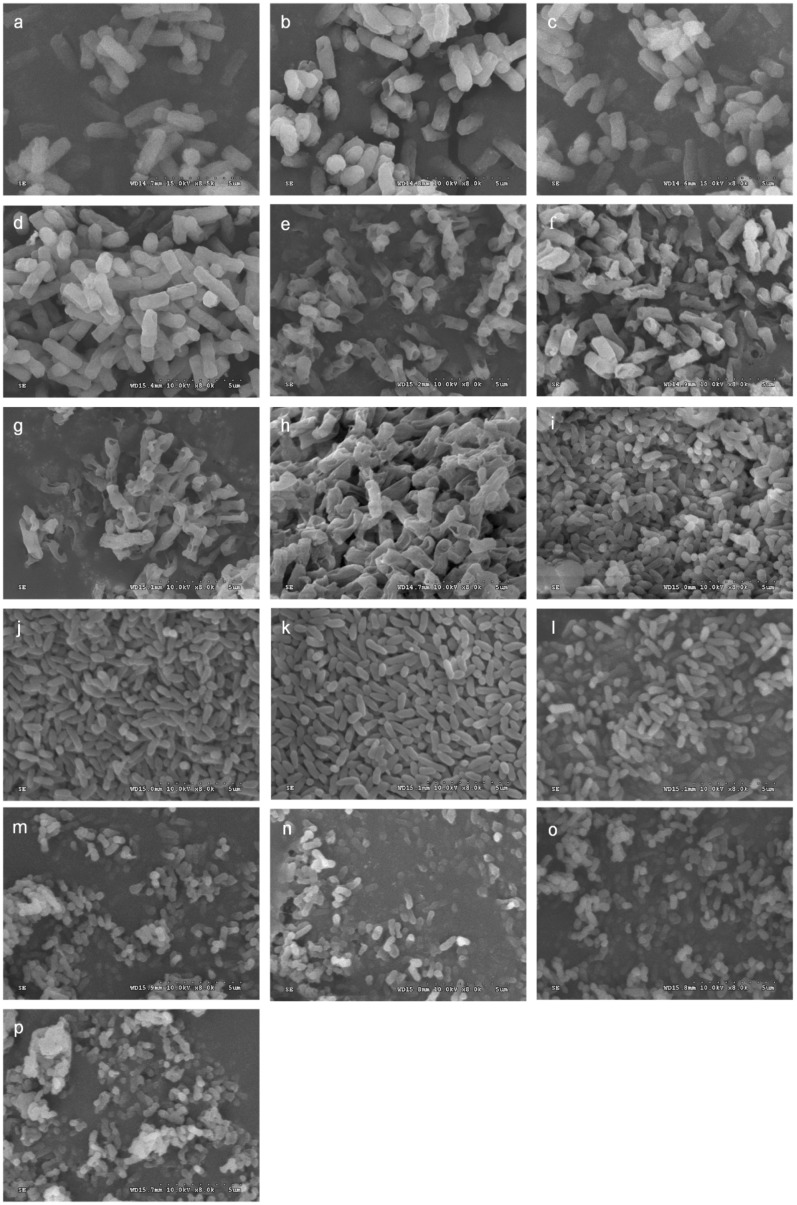
Scanning electron microphotographs of *B. thermosphacta* and *P. fluorescens*. *B. thermosphacta* untreated for 4 h (**a**), untreated for 8 h (**b**), treated with 2% ethanol for 4 h (**c**), treated with 2% ethanol for 8 h (**d**), treated with 3-carene at 1× MIC for 4 h (**e**), treated with 3-carene at 1× MIC for 8 h (**f**), treated with 3-carene at 2× MIC for 4 h (**g**), treated with 3-carene at 2× MIC for 8 h (**h**); *P. fluorescens* untreated for 4 h (**i**), untreated for 8 h (**j**), treated with 2% ethanol for 4 h (**k**), treated with 2% ethanol for 8 h (**l**), treated with 3-carene at 1× MIC for 4 h (**m**), treated with 3-carene at 1× MIC for 8 h (**n**), treated with 3-carene at 2× MIC for 4 h (**o**), treated with 3-carene at 2× MIC for 8 h (**p**).

**Figure 5 molecules-24-03246-f005:**
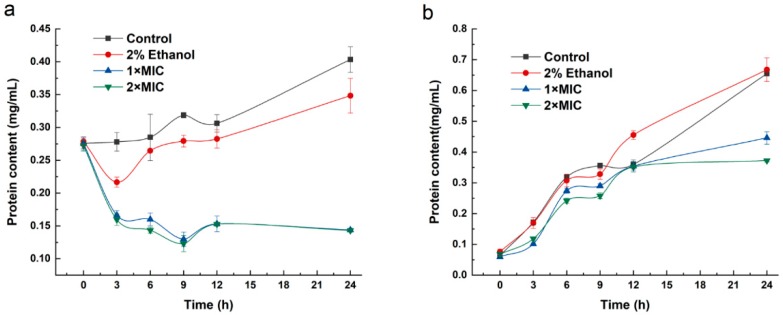
The effect of intracellular protein content after treatment of 3-carene. (**a**) *B. thermosphacta*, (**b**) *P. fluorescens.*

**Figure 6 molecules-24-03246-f006:**
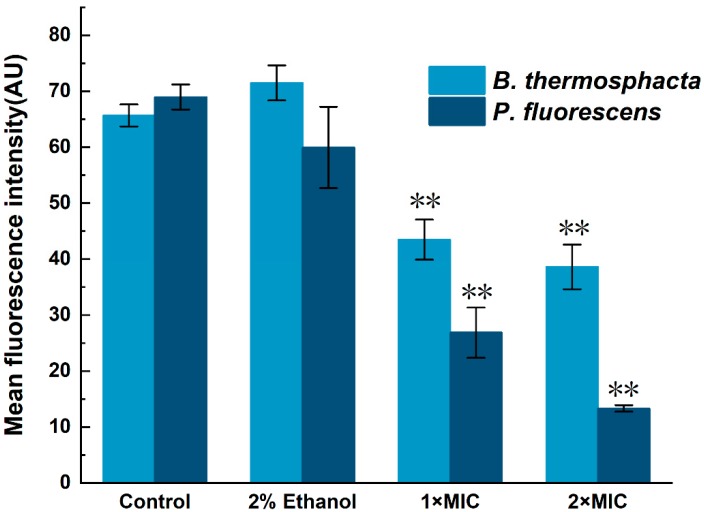
Membrane potential (MP) of (**a**) *B. thermosphacta* and (**b**) *P. fluorescens* treated with 3-carene. The asterisks (**) indicate a significant difference compared with the control at *p* < 0.01.

**Figure 7 molecules-24-03246-f007:**
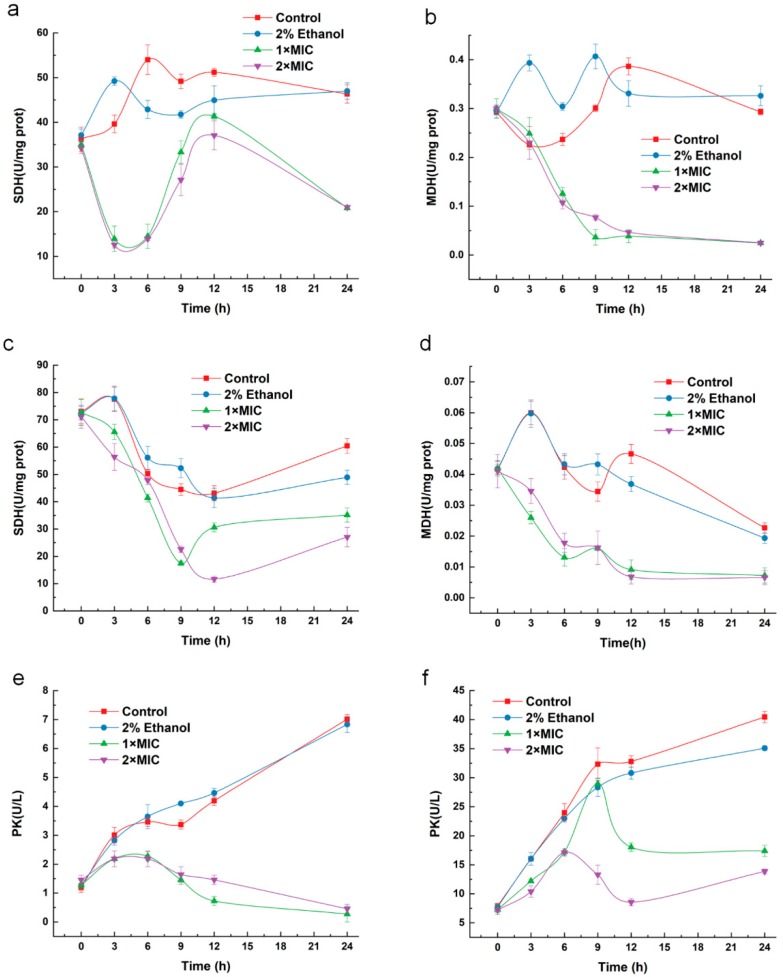
Effects of 3-carene on the activity of succinate dehydrogenase (SDH), malate dehydrogenase (MDH) and pyruvate kinase (PK) in bacteria. (**a**) SDH of *B. thermosphacta*, (**b**) MDH of *B. thermosphacta*; (**c**) SDH of *P. fluorescens*, (**d**) MDH of *P. fluorescens*; (**e**) PK of *B. thermosphacta*, (**f**) PK of *P. fluorescens.*

**Figure 8 molecules-24-03246-f008:**
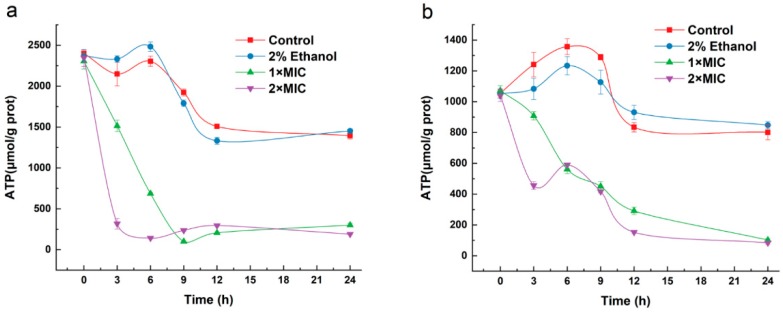
Effects of 3-carene on intracellular ATP in (**a**) *B. thermosphacta* and (**b**) *P. fluorescens.*

**Figure 9 molecules-24-03246-f009:**
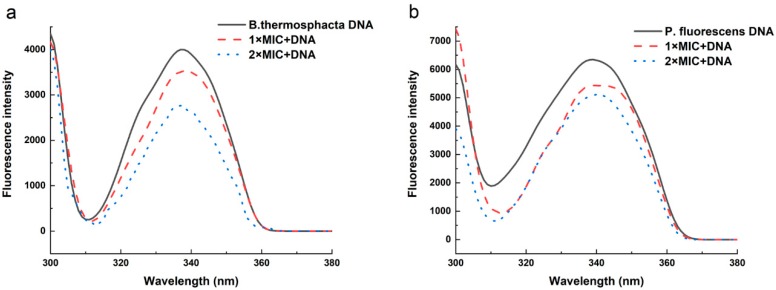
Fluorescence spectra of (**a**) *B. thermosphacta* and (**b**) *P. fluorescens* genomic DNA in the presence of increasing amounts of 3-carene. Fluorescence spectra were measured from 300 to 500 nm (k ex = 280 nm).

**Figure 10 molecules-24-03246-f010:**
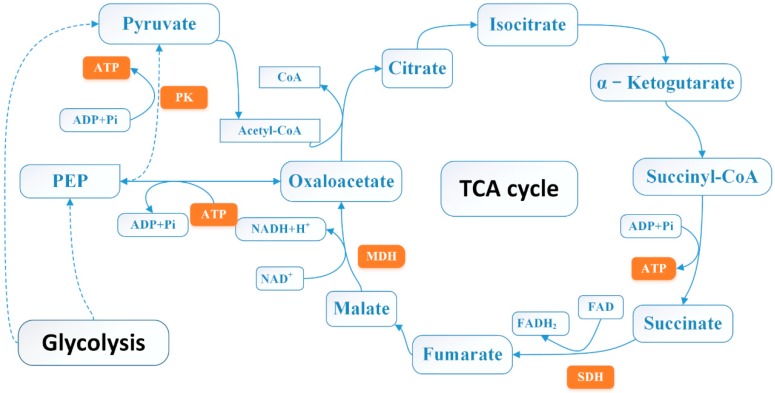
Schematic representation of tricarboxylic acid (TCA) cycle.

**Table 1 molecules-24-03246-t001:** Minimum inhibitory concentration (MIC) of 3-carene against *Brochothrix thermosphacta* and *Pseudomonas fluorescens.*

Bacteria	Control		The Concentration of 3-Carene (mL/L)						
	Water	2% Ethanol	0.625	1.25	2.5	5	10	20	40
*B.thermosphacta*	+++	+++	+++	+++	+++	++	+	−	−
*P. fluorescens*	+++	+++	+++	+++	+++	+++	++	−	−

“−” represents no bacteria; “+” indicates little bacteria; “++” represents a medium number of bacteria; “+++” represents a large number of colonies.

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
