# Peer review of "Antimicrobial Activity and Proposed Action Mechanism of 3-Carene against Brochothrix thermosphacta and Pseudomonas fluorescens"

_molecules, 2019, doi:10.3390/molecules24183246_

Round 1

Reviewer 1 Report

The manuscript entitled "Antimicrobial Activity and Proposed Action Mechanism of 3-Carene against Brochothrix thermosphacta and Pseudomonas fluorescens" is well structured and characterized by a remarkable scientific line in its section.

The abstract contains all the necessary information in a concise and clear way and the graphical abstract is also well designed and enjoyable to see. 

The "introduction" section is a bit poor and could be enriched by works concerning mainly Brochothrix thermosphacta that are available in literature.

All the rest of the manuscript is instead extremely accurate and precise, especially the discussion and the materials and methods.

Therefore, I recommend making these small changes to the introduction, after which the manuscript will undoubtedly be ready for publication.
I also congratulate the Authors on the choice of topic and the way of working demonstrated.

Author Response

Comments and suggestions to the Author

The manuscript entitled "Antimicrobial Activity and Proposed Action Mechanism of 3-Carene against Brochothrix thermosphacta and Pseudomonas fluorescens" is well structured and characterized by a remarkable scientific line in its section. The abstract contains all the necessary information in a concise and clear way and the graphical abstract is also well designed and enjoyable to see. The "introduction" section is a bit poor and could be enriched by works concerning mainly Brochothrix thermosphacta that are available in literature. All the rest of the manuscript is instead extremely accurate and precise, especially the discussion and the materials and methods. Therefore, I recommend making these small changes to the introduction, after which the manuscript will undoubtedly be ready for publication. I also congratulate the Authors on the choice of topic and the way of working demonstrated.

R: We appreciate the reviewer’s positive view of the study and have enriched the introduction section.

B. thermosphacta are Gram-positive and facultative anaerobic bacterium that can cause spoilage in chilled meat and seafood products [3]. Jiang et al. showed that B. thermosphacta are the dominant spoilage bacteria in vacuum-packed pork during chilled storage due to its facultative anaerobic and psychrophilic features, leading to corruption and deterioration [4]. P. fluorescens are Gram-negative bacterium and are commonly found in cold meat, seafood, and dairy products and can cause serious diseases such as septicemia, septic shock and intravascular coagulation [5,6,7].” (line 39-44)

We have studied reviewer’s comments carefully and have made revision which marked in red in the paper. We have tried our best to revise our manuscript according to the comments. Attached please find the revised version.

Reviewer 2 Report

In my opinion, the manuscript has a potential to become a good research paper, however some important comments need to be addressed before the acceptance of the manuscript to be published in Molecules.

Major comments:

Abstract: “3-Carene … occurs naturally in a variety of spices and herbs”. Carene belongs first of all to the main components of essential oils of the conifers, such as Pinus Complete the information. Introduction, line 42: Explain a little more, why “preservatives creates a potential risk to public health” (e.g. give the examples of potential diseases associated with food preservative consumption). Introduction, line 57-58: “3-carene is an effective natural inhibitor” – inhibitor of what? Results, line 71-72: “MIC…was determined by the broth dilution method” – According to the Materials and Methods agar dilution method was used. Additionally, explain why recommended for MIC determination broth dilution method was not used? Results, MIC determination and other results: In my opinion 3-carene MIC value designated for thermosphacta is disputable. According to Table 1 the real MIC against these bacteria is located probably somewhere between 10 and 20 mL/L (even closer 10 mL/L). The concentration of 20 mL/L you used seems to be rather minimum bactericidal concentration (MBC) for B. thermosphacta, what it was confirmed by dynamic grow curves (Fig. 1a). Consider this suggestion to improve (to deepen) the description and the discussion of other results. For instance, results line 93-99, Fig. 2a,b: If 20 and 40 mL/L are rather bactericidal than bacteriostatic concentrations for B. thermosphacta, the lysis of the cells and the release of AKP are a natural consequence of this. Result, line 75-80 and Fig. 1: For thermosphacta you cannot say “The bacteria grew slowly” since there was no growth at all. Results, line 113: “The effect of 3-carene on inhibiting biofilms” - According to the Materials and Methods you used bacterial suspensions, so you did not test the biofilms. Results, line 113-128 and Fig. 3: Except the same comments as above (no 5), describing FDA results additionally take into consideration the effect of 3-carene not only on cell membrane permeability, but also on bacterial metabolic activity. Results, line 212-220 and Fig. 9: Why did you decided to use also 4xMIC to test 3-carene effect on DNA? Since it only one test with expanded range of concentrations, it better to show only MIC and 2xMIC to unify all tests done. It is widely known that essentials oils are quite cytotoxic for eukaryotic cells and cytotoxicity tests are crucial if you want to use such preparations in practice (e.g. as food additive). Have you tested cytotoxicity of 3-carene? If yes, add to the results, if not, perhaps literature data are available on this subject, so add such information to the discussion. Discussion and Conclusions: Consider the suggestion, that probably bactericidal concentrations of 3-carene against thermosphacta de facto was used to improve these subsections. In this context, for instance, the decrease of protein content in the presence of 3-arene may result not only from cell membrane leakage. All methods: The description: “Sterile water and ethanol (2%, v/v) were used as a blank control and negative control, respectively” is confusing, the more that in all figures (results) you have a description: Control, 2% Ethanol. I expect you add water or ethanol (as 3-carene solvent) to bacterial suspensions to check their normal growth / metabolic activity, so untreated bacteria means positive control or just control (as on figures). Blank or negative control means without any bacteria or their products. Explain and improve or complete the description in Materials and Methods.

Minor comments:

Abstract and whole text: Improve Gram-positive, Gram-negative bacteria (start with a capital letter). Abstract, line 16-17: Give full name of thermosphacta and P. fluorescens when the first time in the text. Introduction, line 36: “Food contamination caused by food spoilage and pathogenic bacteria” – The majority of microorganisms, which cause food spoilage are included to facultative pathogens. I advise you to change their status. Introduction, line 56: Improve the sentence: “via reducing biofilm permeability” – Permeability causing leakage of intracellular substances concerns cell membrane / wall, biofilm can be disrupted (release of cells or cell aggregates). Whole text: Talking about microorganisms, plural should rather be used, e.g. thermosphacta are Gram-positive and facultative anaerobic bacteria. Introduction, line 52-53: proper name of “Salmonella typhimurium” is Salmonella enterica enterica serovar typhimurium or Salmonella Typhimurium (serovar written a capital letter, without italics). Introduction, line 53: “Monoterpene is the main component…” - Monoterpenes are … Introduction, line 60-63: Improve the sentence: “The inhibition mechanism …, the inhibition of enzyme activity, the inhibition of nucleic acid synthesis”. Results, Table 1, line 81-83: Enlarge the table or reduce the font size so that the descriptions fit on one line. Results, line 89-90:”AKP…localizes between the cell wall and membrane” – It is better to say in membrane since you are thinking about all organisms, including eukaryotic cells. Result, line 171-172: “the degree at which the MP decreased increased with increasing 3-carene 171 concentration” – Improve the sentence. Discussion, line 223: “ thermosphacta ACCC 03,870” – improve number Discussion, line 229: “[26] showed that serious” - Add first author name et al. before the reference number. Discussion, line 231: “treated with PMFs” - expand the shortcut. Discussion, line 244-248: Improve the fragment “An increase in the amount…to leakage of protoplasmic content” since both sentences contain the same information. Discussion, line 307-308: “[47] showed that flavonoids can effectively inhibit DNA…” - Add first author name et al. before the reference number. Method 4.9, line 391-401: The activity of SDH, MDH and PK can’t be tested using Coomassie Brilliant Blue method – improve description. Add information about assay kits (name, producer) for determination SDH, MDH and PK activity. Method 4.10.: Add information about ATP assay kit (name, producer). References: According to Molecules Instruction for the Authors volume number should be in italics; remove “pp” (e.g. ref. 6),  “vol” (e.g. ref. 19).

Author Response

Dear reviewer,

Thank you very much for your positive and constructive comments and suggestions on our manuscript entitled “Antimicrobial Activity and Proposed Action Mechanism of 3-Carene against Brochothrix thermosphacta and Pseudomonas fluorescens”. (Manuscript ID: molecules-586606).We have studied comments carefully and have made correction according to your suggestion. Revised portion are marked in red in the paper. The main corrections in the manuscript and the responds to the reviewer’s comments are as flowing:

Comments and suggestions to the Author

In my opinion, the manuscript has a potential to become a good research paper, however some important comments need to be addressed before the acceptance of the manuscript to be published in Molecules.

Major comments:

Point 1: Abstract: “3-Carene … occurs naturally in a variety of spices and herbs”. Carene belongs first of all to the main components of essential oils of the conifers, such as Pinus Complete the information.

R: We all think well of your suggestion. We have completed the information in the introduction section according to your suggestion.

“3,7,7-Trimethyl-bicyclo [4,1,0]hept-3-ene (3-carene) is the main components of some conifers and herbs volatile oils, such as Pinus and pepper.” (line 63-64)

Point 2: Introduction, line 42: Explain a little more, why “preservatives creates a potential risk to public health” (e.g. give the examples of potential diseases associated with food preservative consumption).

R: Some detailed information about the potential risk of preservatives to public health has been added in the introduction section.

The improper use of chemical preservatives may cause problems such as carcinogenicity, teratogenicity and residual toxicity.” (line 47-48)

Point 3: Introduction, line 57-58: “3-carene is an effective natural inhibitor” – inhibitor of what?

R: 3-Carene is an effective natural inhibitor of food-borne germs with high content in black and white pepper oil.” And we have added it in our manuscript. (line 64-65)

Point 4: Results, line 71-72: “MIC…was determined by the broth dilution method” – According to the Materials and Methods agar dilution method was used. Additionally, explain why recommended for MIC determination broth dilution method was not used?

R: Thanks for your careful review on our manuscript. The minimum inhibitory concentrations (MIC) of 3-carene against B. thermosphacta and P. fluorescens was determined by the agar dilution method. We have revised this sentence in the article. The broth dilution method was not used for MIC determination since we can’t observe the growth of bacteria well in actual operation.

Point 5: Results, MIC determination and other results: In my opinion 3-carene MIC value designated for thermosphacta is disputable. According to Table 1 the real MIC against these bacteria is located probably somewhere between 10 and 20 mL/L (even closer 10 mL/L). The concentration of 20 mL/L you used seems to be rather minimum bactericidal concentration (MBC) for B. thermosphacta, what it was confirmed by dynamic grow curves (Fig. 1a). Consider this suggestion to improve (to deepen) the description and the discussion of other results. For instance, results line 93-99, Fig. 2a,b: If 20 and 40 mL/L are rather bactericidal than bacteriostatic concentrations for B. thermosphacta, the lysis of the cells and the release of AKP are a natural consequence of this.

R: Thanks for reviewer’s comment on the study. MICs of 3-carene on B. thermosphacta and P. fluorescens were tested using a twofold serial dilution method. We have consulted many references and this twofold serial dilution method is an easy, rapid, and reliable method for the determination of MIC and was widely used in many studies. For example, Yao et al. used a twofold serial dilution method to determined MICs of nobiletin and tangeretin in <Antimicrobial activity of nobiletin and tangeretin against Pseudomonas>. M Chandrasekaran and V Venkatesalu also used this method to determined MIC in <Antibacterial and antifungal activity of Syzygium jambolanum seeds>. This result in the study is reliable. We have improved the sentence to avoid ambiguous as followed; “Briefly, a twofold serial dilution method was used and 3-carene was dissolved in alcohol (20%, v/v) to obtain a concentration of 400 mL/L and serially diluted to 200, 100, 50, 25, 12.5, 6.25 mL/L. A prepared test compound solution (2 mL) was mixed with nutrient agar medium (18 mL) to yield final concentrations of 40, 20, 10, 5, 2.5, 1.25 and 0.625 mL/L. The final concentration of ethanol in the culture medium was maintained at 2% (v/v). Sterile water was added to bacterial suspensions as a blank control and ethanol (2%, v/v) (3-carene solvent) was added to bacterial suspensions as a negative control to check their normal growth. All experiments in this study used this method as a blank and negative control.” (line 334-341) Reviewer's comment on the dynamic grow curves was considerable and this may be due to the strong inhibitory effect of 3-carene on B. thermosphacta during bacterial growth.

Point 6: Result, line 75-80 and Fig. 1: For thermosphacta you cannot say “The bacteria grew slowly” since there was no growth at all.

R: The sentence has been revised as follows; “When 3-carene was added to the medium, B. thermosphacta stopped growing (Fig. 1a) and P. fluorescens grew slowly (Fig. 1b).” (line 84-85)

Point 7: Results, line 113: “The effect of 3-carene on inhibiting biofilms” - According to the Materials and Methods you used bacterial suspensions, so you did not test the biofilms.

R: It has been revised according your suggestion. Section of 2.4 has been changed to “The effect of 3-carene on inhibiting cell membrane of B. thermosphacta and P. fluorescens” (line 121)

Point 8: Results, line 113-128 and Fig. 3: Except the same comments as above (no 5), describing FDA results additionally take into consideration the effect of 3-carene not only on cell membrane permeability, but also on bacterial metabolic activity.

R: The FDA results have been revised according to your suggestion.

“Fluorescein remaining on the cell membrane may be the uptake of FDA and its intracellular hydrolysis by esterase to release free fluorescein [24]. Therefore, this method can also be used as an indicator of cell viability and metabolic activity of bacteria [25].” (line 127-129)

Point 9: Results, line 212-220 and Fig. 9: Why did you decided to use also 4xMIC to test 3-carene effect on DNA? Since it only one test with expanded range of concentrations, it better to show only MIC and 2xMIC to unify all tests done.

R: We all think well of your point. The figure in the manuscript has only showed MIC and 2xMIC to unify all tests done.

Point 10: It is widely known that essentials oils are quite cytotoxic for eukaryotic cells and cytotoxicity tests are crucial if you want to use such preparations in practice (e.g. as food additive). Have you tested cytotoxicity of 3-carene? If yes, add to the results, if not, perhaps literature data are available on this subject, so add such information to the discussion.

R: We all think well of your suggestion and the literature data of cytotoxicity of 3-carene have been added in the discussion.

Lizandra et al. have identified that 3-carene was the main chemical constituents of essential oil of piper cubeba and the significant cytotoxicity was only obtained in the concentration of 200 μg/mL after 24 h treatment. (line 234-236)

Point 11: Discussion and Conclusions: Consider the suggestion, that probably bactericidal concentrations of 3-carene against thermosphacta de facto was used to improve these subsections. In this context, for instance, the decrease of protein content in the presence of 3-carene may result not only from cell membrane leakage.

R: Thanks for reviewer’s advice. The decrease of protein content was supposed due to that 3-carene disrupt protein synthesis, cause metabolic dysfunction.

Point 12: All methods: The description: “Sterile water and ethanol (2%, v/v) were used as a blank control and negative control, respectively” is confusing, the more that in all figures (results) you have a description: Control, 2% Ethanol. I expect you add water or ethanol (as 3-carene solvent) to bacterial suspensions to check their normal growth / metabolic activity, so untreated bacteria means positive control or just control (as on figures). Blank or negative control means without any bacteria or their products. Explain and improve or complete the description in Materials and Methods.

R: Thanks for reviewer’s suggestion. The final concentration of ethanol in the culture medium was maintained at 2% (v/v). Sterile water was added to bacterial suspensions as a blank control and ethanol (2%, v/v) (3-carene solvent) was added to bacterial suspensions as a negative control to check their normal growth. All the description in Materials and Methods has been revised. (line 338-341)

Minor comments:

Point 13: Abstract and whole text: Improve Gram-positive, Gram-negative bacteria (start with a capital letter).

R: The “Gram-positive, Gram-negative bacteria” have been used throughout the manuscript according to your suggestion.

Point 14: Abstract, line 16-17: Give full name of thermosphacta and fluorescens when the first time in the text.

R: It has been revised according to your suggestion. (line 338-341)

Point 15: Introduction, line 36: “Food contamination caused by food spoilage and pathogenic bacteria” – The majority of microorganisms, which cause food spoilage are included to facultative pathogens. I advise you to change their status.

R: We have improved the sentence in the introduction section as per reviewer’s suggestion as follows; “Food contamination caused by facultative pathogens is a serious challenge in the food industry.” (line 37-38)

Point 16: Introduction, line 56: Improve the sentence: “via reducing biofilm permeability” – Permeability causing leakage of intracellular substances concerns cell membrane / wall, biofilm can be disrupted (release of cells or cell aggregates).

R: Thanks for reviewer’s advice. We have improved the sentence in the introduction section as per reviewer’s suggestion as follows; “Thymol, (+) menthol and linalyl acetate are three common monoterpenes that have effective antibacterial activity against the Gram-positive bacterium S. aureus and Gram-negative bacterium E. coli via destroyed cell wall and membrane and causing leakage of intracellular substances.”

Point 17: Whole text: Talking about microorganisms, plural should rather be used, e.g. thermosphacta are Gram-positive and facultative anaerobic bacteria.

R: Plural have been used when talking about microorganisms in the whole manuscript.

Point 18: Introduction, line 52-53: proper name of “Salmonella typhimurium” is Salmonella enterica enterica serovar typhimurium or Salmonella Typhimurium (serovar written a capital letter, without italics).

R: The proper name of “Salmonella Typhimurium” have been used in the manuscript according to your suggestion.

Point 19: Introduction, line 53: “Monoterpene is the main component…” - Monoterpenes are …

R: The sentence has been revised as: “Monoterpenes are the main component of EOs and possesses various biological activities, including antibacterial activity.” (line 59-60)

Point 20: Introduction, line 60-63: Improve the sentence: “The inhibition mechanism …, the inhibition of enzyme activity, the inhibition of nucleic acid synthesis”.

R: The sentence has been changed to ‘The inhibition mechanism of antibacterial compounds has been explored by examining cell wall and membrane permeability damage, protein changes, effecting the enzyme activity and nucleic acid synthesis, etc.” (line 67-69)

Point 21: Results, Table 1, line 81-83: Enlarge the table or reduce the font size so that the descriptions fit on one line.

R: Thanks for reviewer’s suggestion. We have reduced the font size of Table 1, so that the descriptions fit on one line. (line 89)

Point 22: Results, line 89-90: “AKP…localizes between the cell wall and membrane” – It is better to say in membrane since you are thinking about all organisms, including eukaryotic cells.

R: Thanks for reviewer’s advice. We have improved the sentence to remove ambiguous in the result section as followed; “AKP is a kind of intracellular enzyme that is found in many prokaryotes and localizes between the cell wall and membrane.” (line 97-98)  

Point 23: Result, line 171-172: “the degree at which the MP decreased increased with increasing 3-carene 171 concentration” – Improve the sentence.

R: The sentence has been improved as your suggestion.

“In addition, it also revealed that the MFI values of P. fluorescens were lower than those of B. thermosphacta when treated with 3-carene, and the MP decreased while 3-carene concentration increased.” (line 179-181)

Point 24: Discussion, line 223: “thermosphacta ACCC 03,870” – improve number

R: The Strain number have been checked and revised as “B. thermosphacta ACCC 03870 and P. fluorescens ATCC 13525” in the manuscript.

Point 25: Discussion, line 229: “[26] showed that serious” - Add first author name et al. before the reference number. Discussion, line 231: “treated with PMFs” - expand the shortcut.

R: We have added first author name et al. before the reference number. And the shortcut “PMFs” have been expanded as “polymethoxylated flavons”. (line 243)

Point 26: Discussion, line 244-248: Improve the fragment “An increase in the amount…to leakage of protoplasmic content” since both sentences contain the same information.

R: The two sentences have been combined and refined. “An increase in the amount of potassium ion efflux from the cells of B. thermosphacta and P. fluorescens provided further evidence that 3-carene causes damage to the cell wall and membrane, and leading to leakage of protoplasmic content”. (line 257-259)

Point 27: Discussion, line 307-308: “[47] showed that flavonoids can effectively inhibit DNA…” - Add first author name et al. before the reference number.

R: The first author name et al. has been added before the reference number.

Point 28: Method 4.9, line 391-401: The activity of SDH, MDH and PK can’t be tested using Coomassie Brilliant Blue method – improve description. Add information about assay kits (name, producer) for determination SDH, MDH and PK activity. Method 4.10.: Add information about ATP assay kit (name, producer).

R: The description has been improved as followed, “The activities of SDH, MDH and PK were expressed based on the protein concentration which determined by the Coomassie Brilliant Blue method.” (line 406-407) The information about assay kits (JianCheng Bioengineering Institute, Nanjing, China) for determination SDH, MDH and PK activity have been added. (line 410)

Point 29: Method 4.10.: Add information about ATP assay kit (name, producer).

R: Thanks for reviewer’s comment. The information about ATP assay kit (JianCheng Bioengineering Institute, Nanjing, China) has been added. (line 415)

Point 30: References: According to Molecules Instruction for the Authors volume number should be in italics; remove “pp” (e.g. ref. 6), “vol” (e.g. ref. 19).

R: Changes have been made according to Molecules Instruction for the Authors in the reference section. Volume number have been in italics in the reference section. We have checked and removed “vol” in reference. Such as “Xu, J.-G.; Liu, T.; Hu, Q.-P.; Cao, X.-M., Chemical Composition, Antibacterial Properties and Mechanism of Action of Essential Oil from Clove Buds against Staphylococcus aureus. In Molecules, 2016; 21, 1194.” We have checked and remove “pp” in our manuscript. In addition, according to the latest instructions, some Books and Book Chapters are formatted with pp page.

Thank you very much for your comments and suggestions.
